# Creating an Array of Parallel Vortical Optical Needles

**Paulius Šlevas [†] and Sergej Orlov *,[†]**

Center for Physical Sciences and Technology, Coherent Optics Laboratory, Sauletekio Ave. 3,
LT-10257 Vilnius, Lithuania; paulius.slevas@ftmc.lt

**\*** Correspondence: sergejus.orlovas@ftmc.lt

**[†]** These authors contributed equally to this work.

**Abstract:** We propose a method for creating parallel Bessel-like vortical optical needles with an arbitrary axial intensity distribution via the superposition of different cone-angle Bessel vortices. We analyzed the interplay between the separation of individual optical vortical needles and their respective lengths and introduce a super-Gaussian function as their axial profile. We also analyzed the physical limitations to observe well-separated optical needles, as they are influenced by the mutual interference of the individual beams. To verify our theoretical and numerical results, we generated controllable spatial arrays of individual Bessel beams with various numbers and spatial separations by altering the spectrum of the incoming laser beam via the spatial light modulator. We demonstrate experimentally how to implement such beams using a diffractive mask. The presented method facilitates the creation of diverse spatial intensity distributions in three dimensions, potentially finding applications in specific microfabrication tasks or other contexts. These beams may have benefits in laser material processing applications such as nanochannel machining, glass via production, modification of glass refractive indices, and glass dicing.

**Keywords:** diffraction; optical engineering; spatial light modulator; Bessel beam; optical vortex; optical needle; focal line; translation of Bessel beams





## 1. Introduction

Over time, lasers have proven to be a versatile tool across various domains, including manufacturing [1], biomedicine [2,3], telecommunication [4,5], and more. Precise adjustment of laser parameters is crucial for achieving specific applications, with the beam shape being a critical factor. The Gaussian beam is the most-common due to its ability to be focused to a small volume.

The structure and properties of the electromagnetic field play a vital role in modern optics and its applications [6–9]. However, non-diffracting beams have recently gained traction due to their capacity to maintain shape over long distances compared to Gaussian beams [8]. One example of structured illumination is non-diffracting beams, such as Bessel beams [10,11], known for their elongated focal line and self-reconstruction capability [12,13]. Laser beams with a long depth of focus and a narrow transverse intensity distribution [14] are useful for laser applications. These beams can be generated by axicons [15,16]. They have shown benefits in laser material processing applications like nanochannel machining [17], glass via production [18], modification of glass refractive indices [19], and glass dicing [20–22]. In other fields, they have been used successfully for THz imaging [23], particle manipulation [24], and microscopy [25].

They have a fixed bell-shaped axial intensity distribution in the Bessel zone [26]. However, this distribution may not be optimal for some applications, such as laser micromachining [17,27–30] and optical tweezers [31]. For some applications, such as surface patterning, Bessel beams can provide a less-sensitive process to surface height variations [32–34], but shaping the focal region could improve stability even further. Several methods to create non-diffractive beams with a flat longitudinal intensity distribution

have been suggested, for example the implementation of a logarithmic axicon [35,36], modulating the spatial spectrum of the Bessel beam [37,38], or using diffractive optical elements [29,39].

Similar characteristics are exhibited by other types of structured light without diffraction, such as Mathieu-Gaussian [40] and parabolic-Gaussian (Weber-Gaussian) beams [41,42]. Another well-known non-diffracting beam is the Airy beam [43], notable for its main lobe propagating along a parabolic trajectory in the longitudinal plane [44]. Airy beams have been used for particle trapping, optical manipulation [45–47], machining curved structures along the beam path [48], multiphoton polymerization [49], and imaging around obstacles in the THz wavelength [50].

In some applications, additional engineering of the focal zone is desirable. The axial behavior of the laser beam, another aspect of beam shaping, has been rapidly evolving [12,51,52] since the introduction of non-diffracting Bessel beams [11], which have a large ratio between the transverse and longitudinal widths of the focal line [10]. The discovery of axial engineering has further stimulated research on focal line engineering, leading to the development of optical needles [38,53–55].

Another degree of freedom is associated with the azimuthal phase dependence of higher-order non-diffracting Bessel beams $J_m$, expressed as $\exp(im\phi)$, where $m$ is a topological charge. This dependence results in a phase singularity, also known as an optical vortex [56–58], which presents as a helical wavefront with $m$ nested screw dislocations [59,60]. Optical vortices, commonly observed in electromagnetics, carry orbital angular momentum [61]. The first instance of optical engineering using cylindrical beams was reported in Ref. [62], where a coherent superposition of the Bessel and Neumann beams was considered, despite the Neumann beam having an amplitude singularity on the axis. By implying the periodicity of the transverse beam pattern in the infinite sum of Bessel eigenmodes, rotating patterns were realized using binary diffractive elements [63,64]. An alternative approach to generating rotating scale-invariant fields was presented in [65].

A single vortex is a dark spot of light with a phase singularity at its core. Consequently, its three-dimensional trajectory can be considered as a unified entity, especially considering that time reversal would change the signs of single vortices in the beam. Therefore, a negatively charged optical vortex can be interpreted as a positively charged vortex propagating backwards, and the creation and annihilation of positive and negative vortices in pairs under diffraction can be viewed as a dark knot of light [66–68]. This interpretation of the intricate evolution of vortical structures with various topologies has led to the emergence of terminology such as knots, braids, and bundles of dark light [69].

Multispot beam splitting is another attractive topic in the field. Various techniques are used to create an array of Gaussian beams, for instance direct laser interference [70,71], beam shaping by a free-form surface lens array [72], by an SLM and a phase mask created by iterative algorithms [73], or by diffractive optical elements [74,75]. The generation of a Bessel beam array has also been reported in the literature. The superposition of multiple shifted axicon holograms in an SLM [76] or the illumination of the axicon with a phase-modulated beam [77] can lead to the formation of multiple Bessel beams. A particular microstructure array can also be used for this purpose [78]. Furthermore, when using these elaborate intensity distributions, aberrations can have positive effects on these beams [79] or can suffer negative aberrations when they pass through a planar dielectric material interface (e.g., air–bulk material) [38,55,80], which affects their performance and efficiency and requires correction [81].

In this work, we propose a method for creating parallel Bessel-like optical needles with adjustable individual axial intensity patterns and individual topological charges. We used a superposition of vortical Bessel beams with specific axicon angles and complex amplitudes, as suggested in Refs. [37,38,77,82–84]. We also demonstrate experimentally how to implement such beams using a spatial light modulator. We studied the interaction of the topological charge and transverse displacements of individual vortical Bessel-like needles of different topological charges. This investigation is important for practical

applications, as it improves the laser energy deposition and the axial intensity distribution of the optical needles [77,85].

## 2. Axial Intensity Control in an Vortical Optical Needle and Translation of Optical Needles

In this section, we outline the theoretical foundation for generating arrays of parallel Bessel-like vortical optical needle beams characterized by a controlled axial intensity pattern. Our approach involves employing ideal Bessel vortices as fundamental basis functions, derived from an angular spectrum described by Dirac's delta function [86]. These non-diffracting beams serve as sufficiently accurate approximations of intensity distributions experimentally observed in the vicinity of the Fourier lens's focal point.

### 2.1. Axial Intensity Control in an Optical Needle

A non-diffracting Bessel vortex is a solution of the scalar Helmholtz equation in the cylindrical coordinates:

$$\psi(\rho, \varphi, z) = J_m(k_\rho \rho) \exp(im\varphi + izk_z), \tag{1}$$

where $\psi$ is the electric field, $J_m$ is the $m$-th order Bessel function, $\rho, \varphi, z$ are cylindrical coordinates, $m$ is the topological charge, and $k_\rho$, $k_z$ are the radial and longitudinal components of the wavevector $\mathbf{k} = (k_\rho, 0, k_z)$. The components of the independent wave vectors are defined through the cone angle $\theta$ of the Bessel beam as $k_\rho = k \sin\theta$ and $k_z = k \cos\theta$ [86].

Every solution to the scalar Helmholtz equation can be expressed through a 3D integral comprising plane waves with distinct wave vectors. This integral can be simplified to a 2D form when dealing with axisymmetric fields, as outlined by Stratton [86]. This two-dimensional representation is based on a Fourier–Bessel transform and can be modified for our specific needs by adjusting the integration variables from $k_\rho$ to $k_z$. The axial component of the wave vector becomes the coordinate of the Fourier transform within the integral expression.

A comprehensive discussion on this method for designing axial profiles can be found in extensive detail in [37,38,82]. When the radial coordinate $\rho$ is zero, the expression of the combined electric field becomes a Fourier series with respect to the axial coordinate $z$, achieved through the superposition of the Bessel beams, see Equation (1). In this context, a Fourier integral is explicitly defined as

$$\Psi(\mathbf{r}) = \int_{-\infty}^{\infty} A(K_z + k_{z0})\psi(\mathbf{r}; K_z)dK_z \tag{2}$$

where $k_z = k_{z0} + K_z$, where $k_{z0}$ is a carrier wave vector and $A(k_z)$ is the spatial spectral amplitude of the combined beam. The carrier wave vector $k_{z0}$ restricts $K_z$ to forward propagating waves only by shifting the spectral coordinates $k_z$ to positive values. A specific case of $m = 0$ was discussed in Ref. [55]; here, we investigated the general case. Near the optical axis $\rho \approx 0$, we can rewrite Equation (1) as

$$\psi(\rho, \varphi, z) = \frac{1}{m!}\left(\frac{k_\rho \rho}{2}\right)^m \exp(im\varphi + izk_z), \tag{3}$$

so, using Equation (3) in Equation (2) results in

$$\Psi(\mathbf{r}) = \frac{1}{m!}\left(\frac{k_\rho \rho}{2}\right)^m \exp(im\varphi + izk_{z0}) \int_{-\infty}^{\infty} A(K_z + k_{z0})\exp(izK_z)dK_z \tag{4}$$

Thus, finally, we arrive at the expression:

$$\Psi(\mathbf{r}) = \frac{1}{m!}\left(\frac{k_\rho \rho}{2}\right)^m \exp(im\varphi + izk_{z0})f(z), \tag{5}$$

where $f(z)$ is the axial distribution of the combined vortical beam with the engineered axial profile, and this profile is expressed as

$$f(z) = \int_{-\infty}^{\infty} A(K_z + k_{z0}) \exp(izK_z) dK_z. \tag{6}$$

Alternatively, knowing the axial profile $f(z)$ of the vortical optical needle, we can arrive at the spatial spectra $A(k_z)$ as

$$A(k_z) = \frac{1}{2\pi} \int_{-\infty}^{+\infty} f(z) e^{-iK_z z} dz. \tag{7}$$

Therefore, the continuous superposition of Bessel beams $\Psi(\mathbf{r})$, as defined in Equation (2) with a spatial spectrum (Equation (3)), is capable of axial intensity properties resembling those defined by a function $f(z)$.

An optical needle (or an optical bottle) is created when the axial profile $f(z)$ is a constant step function. The step function with abrupt start and ending points will cause undesired effects due to the Gibbs phenomenon. Therefore, our aim is to mimic this desired profile by selecting the function $f(z)$ to be a super-Gaussian function:

$$f(z) = \exp\left[-\left(\frac{z - z_0}{z_0}\right)^{2N}\right] \tag{8}$$

where $z_0 = L / \left[2 \log\left(2/2^{1/2N}\right)\right]$ is a some control parameter of the profile at the full-width at half-maximum (FWHM), $N$ is the order of a super-Gaussian beam, and $L$ is the length at the FWHM. This choice enables smoother axial intensity profiles; see Figure 1.

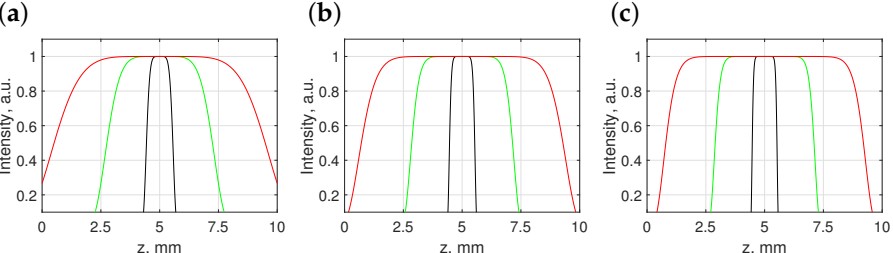

**Figure 1.** Examples of axial profiles used for both numerical simulations and experimental measurements; the order of the super-Gaussian function is $N = 3, 5, 7$ in (**a**–**c**), respectively. The lengths $L$ of the axial profile function $f(z)$ are 1 mm (black), 4 mm (green), and 8 mm (red).

### 2.2. Translation of the Vortical Optical Needle

The control of the transverse position of an optical needle can be performed with the help of the addition theorem of Bessel beams [86]:

$$J_n\left(k_\rho \rho_2\right) e^{in\varphi_2} = \sum_{m=-\infty}^{\infty} J_m\left(k_\rho \rho_{12}\right) J_{n+m}\left(k_\rho \rho_1\right) e^{im(\varphi_1 - \varphi_{12})}. \tag{9}$$

Here, the indices 1 and 2 indicate the coordinates of the first and second cylindrical coordinate systems, $n$ is an index, and $\rho_{12}$ and $\varphi_{12}$ are the coordinates of the translated origin $O_1$ with respect to the untranslated origin $O$; see Figure 2. The spatial spectrum of individual Bessel beam (see Equation (1)) is the Dirac delta function multiplied by the azimuthal phase term; therefore, the shifted coordinates becomes

$$\hat{\psi}(k_\rho, \varphi_k) = \sum_{m=-\infty}^{\infty} J_m\left(k_\rho \rho_{12}\right) e^{-im\varphi_{12}} \frac{i^m e^{im\varphi_k} \delta\left(k \sin\theta - k_\rho\right)}{k_\rho}, \tag{10}$$

where $\varphi_k$ is the azimuth of the spatial spectral coordinates. In this way, the Bessel beam with the origin at the shifted point is expressed as a superposition of Bessel beams in the non-shifted origin; see Figure 2.

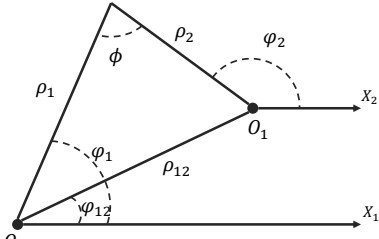

**Figure 2.** Depiction of the original and translated cylindrical coordinates. $O$ is the original coordinate $(x_1, \rho_1, \varphi_1)$, and $O_1$ is the center of the translated coordinate $(x_2, \rho_2, \varphi_2)$.

Our intention here is to have a number $p = 1, 2, \ldots, P$ of independent parallel vortical needles, each with its own axial profile $f_p(z)$ and individual position $x_p, y_p$ in the transverse plane. According to the superposition principle, the resulting spatial spectrum of the combined beam is

$$\hat{\Psi}(k_x, k_y) = \sum_p \int_{-\infty}^{\infty} A_p(K_z + k_{z0})\hat{\psi}_p(k_x, k_y)dK_z. \tag{11}$$

where

$$A_p(k_z) = \frac{1}{2\pi} \int_{-\infty}^{+\infty} f_p(z)e^{-iK_z z}dz. \tag{12}$$

and $\hat{\psi}_p$ is given in Equation (10) for translation to the coordinate center located at $x_p, y_p$.

### 3. Experimental Results

*3.1. Optical Setup*

The optical setup used throughout the experiment is displayed in Figure 3. A $\lambda = 532$ nm DPSS laser beam is attenuated by a half-waveplate (HWP) and a Brewster's polarizer (BP) and, then, expanded by $40\times$, with an $NA = 0.65$ microscope objective (MO1), and the beam is collimated by the lens L1 ($f = 125$ mm). The beam is then directed to a spatial light modulator (SLM) (PLUTOVIS-006-A, HOLOEYE Photonics AG, 8 bit, $1920 \times 1080$ pixels, 8 µm pitch) by means of a non-polarizing beam splitter (BS) at a 0 deg angle. The correct polarization direction is set by the polarizer (BP). The size of the beam (~8.5 mm diameter) is selected by an aperture (A1) to fill the SLM screen (8.64 mm). The SLM shapes the beam spectrum, which is directed to a $3\times$ demagnifying optical setup $4f$ comprised of lenses L2 ($f = 150$ mm) and L3 ($f = 50$ mm). This demagnifies the transverse coordinates three times and the longitudinal coordinates nine times. In the focal plane of lens L2, additional diffraction maxima created by the checkerboard mask are filtered out by aperture A2. Lastly, the spectrum is focused by a Fourier lens L4 ($f = 25.4$ mm). The dimensions of the central lobe would be too small to register directly by a camera; therefore, the beam is captured by an optical imaging setup assembled on a motorized stage. This setup contains a $10\times$ microscope objective (MO2), a collimating lens L5 ($f = 200$ mm), and a CCD camera. The entire beam is scanned using 0.05 mm steps.

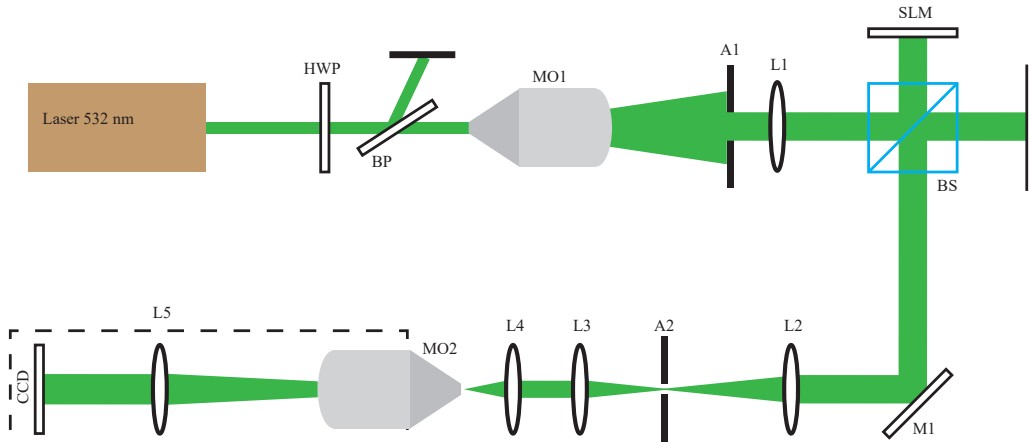

**Figure 3.** Optical setup of the experiments. HWP is the half-wave plate; BP is the Brewster polarizer; MO1, MO2 are the microscope objectives; L1, L2, L3, L4, and L5 are the lenses; A1 and A2 are the apertures; SLM is a spatial light modulator; BS is a beam splitter; M1 is a mirror; CCD is a charge-coupled device camera.

*3.2. Creation of Single Vortical Needles*

The optical setup used in this work is a bit different from that used in our previous research [38,55]. This time, a different set of lenses was used for magnification. Therefore, we begin with experiments to verify the concept of single vortical needles with controllable axial profiles $f(z)$, described in Equation (8). The spatial spectra of the engineered beams are found using Equation (7) and encoded in a phase-only mask using a checkerboard method [87]. The parameters for the axial profiles $f(z)$ used in the experiments were the same as in the theoretical section; see Figure 1.

Our intention here is to reproduce optical needles without topological charge (i.e. $m = 0$) to ensure the smooth operation of the optical setup before proceeding to the masks with topological charges. For this purpose, we selected three different lengths of optical needles $L = 1$ mm, $L = 4$ mm, and $L = 8$ mm and experimentally verified their creation (see Figure 4). We performed experiments with various orders $N$ of the super-Gaussian function, but our experimentation identified the number $N = 7$ as optimal, which is in line with previous work [38,55].

The optical needle with the shortest length $L = 1$ mm has a central spike that is ~7.3 µm in diameter (at the intensity level $1/e^2$) and two detectable rings in the transverse intensity pattern; see Figure 4a. This is an expected outcome given its small longitudinal dimension. As the length $L$ decreases, the properties of the optical needle are not similar to the properties of a Bessel beam, but are more like those of a Gaussian beam. Moving on to the four-time longer optical needle with $L = 4$ mm, we observed the appearance of additional rings (seven in total) around the central spike. The size of the central spike remained the same within the experimental tolerance, and the transverse profile largely resembles a Bessel-Gaussian beam with a significant number of concentric rings surrounding the center of the beam. Lastly, we increased the length of the optical needle twice more to $L = 8$ mm; see Figure 4c. The system of concentric rings becomes more pronounced, and the central lobe is not significantly affected. We verified the propagation of these optical needles by measuring the intensity of the central lobes for various positions of the $z$ coordinate; see Figure 4d. In general, the behavior was detected to be as expected from numerical simulations, with no sharp oscillations on the edges; the intensity drop was smooth, as desired. However, we did observe some axial oscillations, which might be caused by inaccuracies in the positioning of the translation stage and some possible misalignment in the optical system.

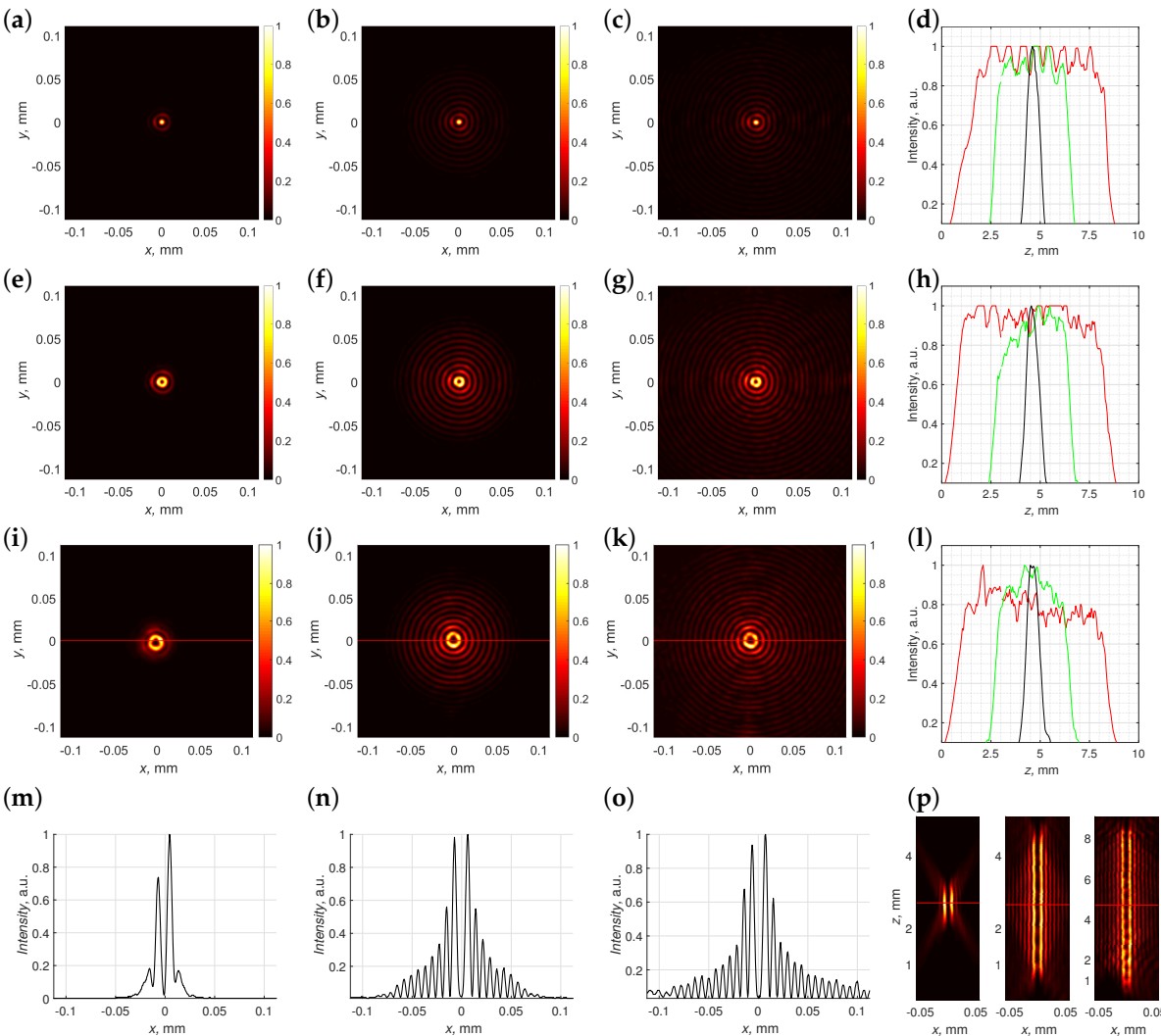

**Figure 4.** Transverse intensity distributions for vortical needles with topological charges $m = 0$ (**a–c**), $m = 1$ (**e–g**) and $m = 2$ (**i–k**) and lengths of the axial super-Gaussian profile $L = 1$ mm (**a**,**e**,**i**), $L = 4$ mm (**b**,**f**,**j**), and $L = 8$ mm (**c**,**g**,**k**). Longitudinal axial profiles of vortical optical needles, measured in the brightest ring (or central lobe) for topological charges $m = 0$ (**d**), $m = 1$ (**h**), and $m = 2$ (**l**) and different lengths $L = 8$ mm (red), $L = 4$ mm (green), and $L = 1$ mm (black). Intensity distributions (**m–o**) of cross-sections marked by a red line in (**i–k**), respectively. Longitudinal intensity distributions for optical needles with topological charge $m = 2$ (**p**).

Having verified that the optical setup acts at an intended level of performance, we now introduce nonzero topological charges $m = 1$ and $m = 2$; see Figure 4. Starting with the shortest vortical needle with a length of $L = 1$ mm, we observed similarities with the previous case; compare Figure 4e,i to Figure 4a. We observed two pronounced rings, the first ring with the vortical core inside and the second one surrounding it. The third ring was weak in both cases. In the expected manner, the radii of the first rings were different: the higher topological charge resulted in a larger central ring; compare Figure 4e to Figure 4i. In the case of the topological charge $m = 1$, the size of a dark spot inside the first ring was ~5.6 μm, measured at a $(1 - 1/e^2)$ intensity level. Setting the length of the axial profile to four-times larger values $L = 4$ mm immediately resulted in the appearance of a good pronounced concentric structure with nine rings in it for both topological charges. The sizes of the central rings surrounding the vortex cores with topological charges $m = 1$ (Figure 4f) and $m = 2$ (Figure 4j) did not change significantly. Lastly, setting the length of the super-Gaussian axial profile to $L = 8$ mm gave us the transverse intensity patterns depicted

in Figure 4g,k. In a similar fashion to the non-vortical optical needle (see Figure 4c), the ring-like structure of the field became more pronounced. We verified the intended action of the phase mask by measuring the intensity on the first ring while performing a $z$ scan; see Figure 4h,l. In both cases, the axial profile of the vortical beams with the shortest length $L = 1$ mm resembled our expectation well; see the black curves in Figure 4h,l. Longer axial profiles had the expected lengths, but were somehow distorted; see the green curves in Figure 4h,l. This might happen due to the azimuthal intensity fluctuations on the first ring; compare to Figure 4 f,j. We might have used a non-optimal detection method or some misalignment was present in the optical setup. The situation improved for axial profiles designed with length $L = 8$ mm; see the red curves in Figure 4h,l. For the topological charge $m = 1$, we were able to measure the intended axial profile. The axial profile of the topological charge $m = 2$ was flat enough, but some spikes appeared. As we did not integrate azimuths into a ring, but measured them at a single azimuthal angle, this might have occurred due to the coherent addition of a small background, which caused splitting of the central vortex and the appearance of single charged vortices [88,89]. Figure 4m–o show the cross-sections of the $m = 2$ beams marked by a red line in Figure 4i–k, respectively.

As stated above, the main ring was intensity dominant for the shortest optical needle. The first side ring was less than 20% of the maximum (Figure 4). Side rings appeared with the increasing length of the optical needle. For the case of $L = 4$ mm, the first side ring was 55% while for the case of $L = 8$ mm, it was 65%. Both of these values were higher compared to the second ring intensity of the ideal second-order Bessel beam, which would be 42% of the maximum. The size of the dark central spot was ~11.2 µm, which was twice as large compared to the intensity minima of the vortical optical needle of topological charge $m = 1$. Lastly, in Figure 4p, we present the $xz$ distributions of the optical needles of lengths $L = 1$, $L = 4$, and $L = 8$ mm and with topological charge $m = 2$. Smooth intensity distributions were generated for optical needles with $L = 1$ mm and $L = 4$ mm. In the case of $L = 8$ mm, axial modulation was present, which might have occurred due to the splitting of the central vortex into single charged vortices [88,89], as mentioned before.

### 3.3. Creation of an Array of Optical Needles

Having ensured ourselves that single vortical Bessel-like beams with a controllable axial profile can be created similarly as optical vortices, we proceeded to the experimental verification of the creation of an array of independent optical needles using Equation (11). In our previous study, we already investigated this question (see Ref. [55]), but as the optical setup differed from that used in those studies, we will briefly verify that those results are still valid under different conditions.

For this purpose, we created four different phase masks for a set of three optical needles with topological charges $m = 0$, angles $\varphi_{12} = -\pi/4$, and $\varphi_{13} = 3\pi/4$ having four distinct values of $\rho_{12} = \rho_{13}$; see Figure 5. In the first case, we wished to create an array where two adjacent optical needles are separated by $\rho_{12} = 20\lambda$; see Figure 5a. We observed a homogeneous rod-shaped structure, which appeared in the combined beam, with a transverse length similar to the combined distance $\rho_{12} + \rho_{13} = 40\lambda$. However, the side lobes in the direction perpendicular to the axis of the array were comparatively strong, making the spatial lobe of the combined beam complex. This was an expected outcome, as we estimated $f/\# \approx 9$. This implies a classical diffraction limit of approximately 11.5 µm or approximately $22.1\lambda$, so the individual objects are not resolvable in Figure 5a. Moving on to the next case, when $\rho_{12} = \rho_{13} = 40\lambda$ (see Figure 5b), we observed a separation of the rod-shaped structure in the center of the array into three distinct lobes, located in the engineered positions and separated by less-intense rod-shaped intensity patterns. This is a minimal separation distance, which we detected to have a distinct three-spike pattern. A further increase in the separation parameter to $\rho_{12} = \rho_{13} = 60\lambda$ resulted in visibly enhanced central lobes of individual optical needles and reduced the intensity of the side lobes; see Figure 5c. Most notably, though individual objects had only one to two intense rings in the transverse plane (see Figure 4a), the structure of the side lobe appeared to be spatially larger

than that of a single needle. Moving on to even larger spatial separation $\rho_{12} = \rho_{13} = 80\lambda$ resulted in a clearly pronounced array of three optical needles with additional interference patterns; see Figure 5d.

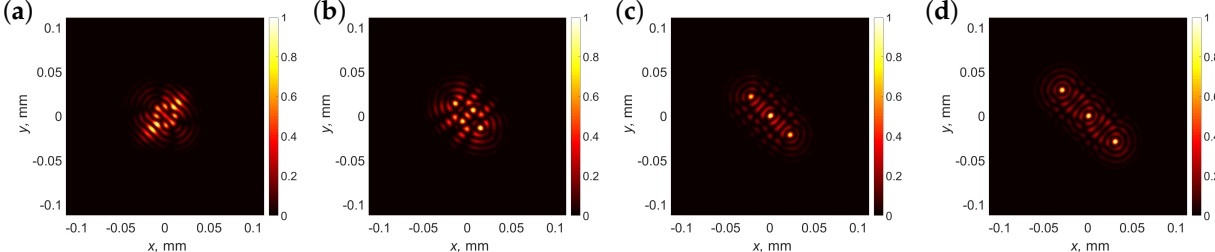

**Figure 5.** Transverse intensity distributions for an array of three optical needles with topological charges $m = 0$ and the length of the axial super-Gaussian profile $L = 1$ mm. The spatial separation of the individual optical needles is $\rho_{12} = 20\lambda$ (**a**), $\rho_{12} = 40\lambda$ (**b**), $\rho_{12} = 60\lambda$ (**c**), and $\rho_{12} = 80\lambda$ (**d**).

Thus, we have confirmed that the optical setup in use acted as intended.

### 3.4. Creation of an Array of Vortical Needles with Individual Topological Charges

In this section, we introduce into the array individual topological charges $m$. We start with a particular example of three optical vortical needles with topological charges $m_1 = 1$, $m_2 = 2$, and $m_3 = 0$. We investigated how the lengths $L$ of individual super-Gaussian axial profiles together with separation distances $\rho_{12}$ and $\rho_{13}$ influence the resulting intensity profile of the combined beam; see Figure 6.

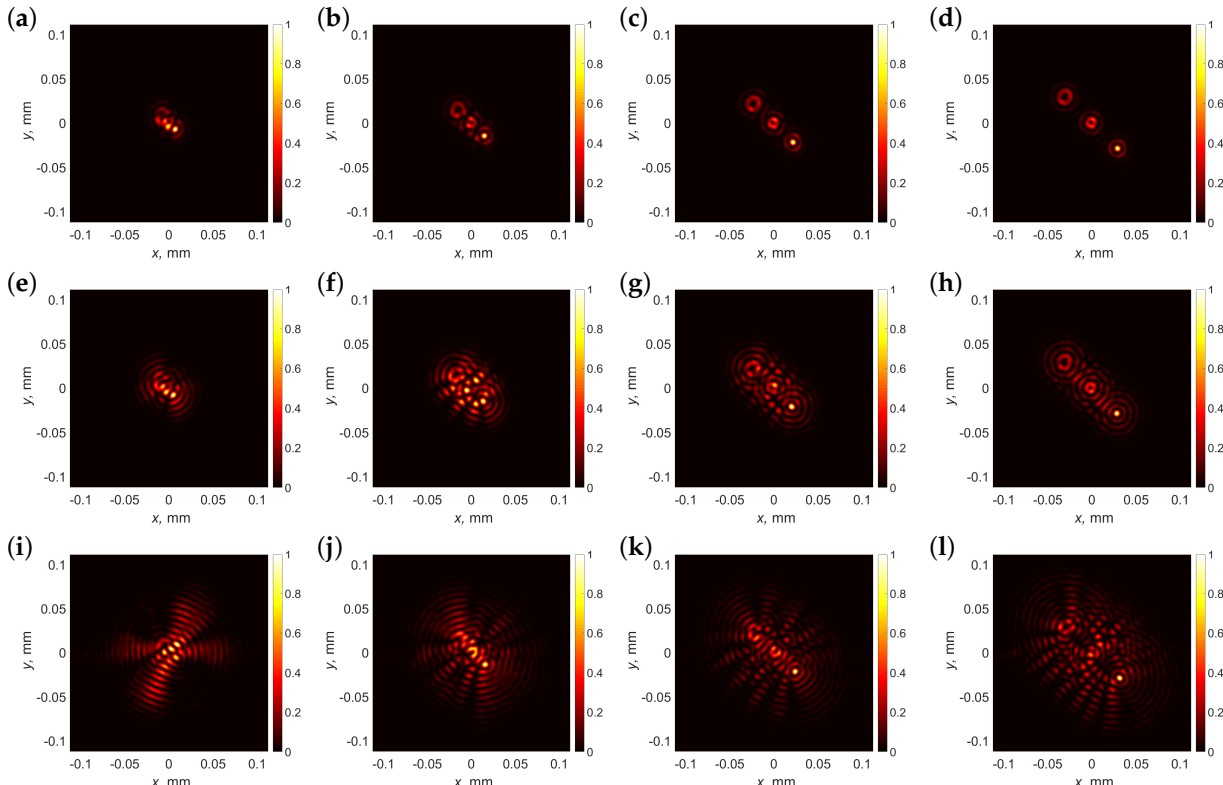

**Figure 6.** Transverse intensity distributions for an array of three optical needles with topological charges $m_1 = 1$, $m_2 = 2$, and $m_3 = 0$ and the length of the axial super-Gaussian profile $L = 1$ mm (**a**–**d**), $L = 2$ mm (**e**–**h**), and $L = 4$ mm (**i**–**l**). The spatial separation of individual optical needles is $\rho_{12} = 20\lambda$ (**a**,**e**,**i**), $\rho_{12} = 40\lambda$ (**b**,**f**,**j**), $\rho_{12} = 60\lambda$ (**c**,**g**,**k**), and $\rho_{12} = 80\lambda$ (**d**,**h**,**l**).

The first case that we studied was the case where the azimuthal locations of adjacent vortical structures are given by the angles $\varphi_{12} = -\pi/4$ and $\varphi_{13} = 3\pi/4$. The separation distances were $\rho_{12} = \rho_{13}$ in all cases. We note that the interference pattern of overlapping Laguerre-Gaussian vortices was studied in Ref. [90], so in the case of coaxial Bessel vortices, we should expect some similar effects. In the first case, we wished to create an array where two adjacent optical needles are separated by $\rho_{12} = 20\lambda$; see Figure 6a,e,i. In general, we see that, for all axial profile lengths $L$, the introduction of vorticity resulted in a largely skewed and distorted profile, although needles with charges $m_3 = 0$ and $m_2 = 2$ can be recognized in the profile of the combined beam for the smallest length $L = 1$ mm. Longer structures were unrecognizable; see Figure 6e,i.

As a next step, we increased the separation lengths $\rho_{12} = \rho_{13}$ to $40\lambda$; see Figure 6b,f,j. For the case with the shortest lengths $L = 1$ mm, we observed an almost complete separation of individual profiles; see Figure 6b. The increase in the length of the optical vortical needle distorted the combined beam; see Figure 6f,j. A further increase in the separation of individual optical beams to $\rho_{12} = \rho_{13} = 60\lambda$ is shown in Figure 6c,g,k. We see that, for $L = 1$ mm, the individual transverse profiles were now fully separated; see Figure 6c. A twofold increase in the length of individual objects was induced as multiple interference patterns adjacent to the individual center rings, but the structure was recognizable and looked as expected; see Figure 6g. A further twofold increase in the length of the optical needles induced stronger interference distortions (see Figure 6k), and as expected, the central optical needle was the most-distorted. Lastly, we further increased the separation distance to $\rho_{12} = \rho_{13} = 80\lambda$; see Figure 6d,h,l. The first two cases were distinctly pronounced (see Figure 6d,h), while the array of the three longest needles showed the main distorted features of the central and upper structure in the transverse plane (see Figure 6l).

Lastly, we were curious whether the order of the topological charges in the combined array had any influence on the quality of the combined beam; see Figure 7. For this investigation, we chose an intermediate length of an individual structure $L = 2$ mm, and the individual topological charges were now $m_1 = 2$, $m_2 = 1$, and $m_3 = 0$. The general idea behind this experiment was that the spatial extent of the higher-order vortex was larger, so the change in the spatial position of the vortex with topological charge $m = 2$ will have an immediate effect on the intensity pattern of the combined beam.

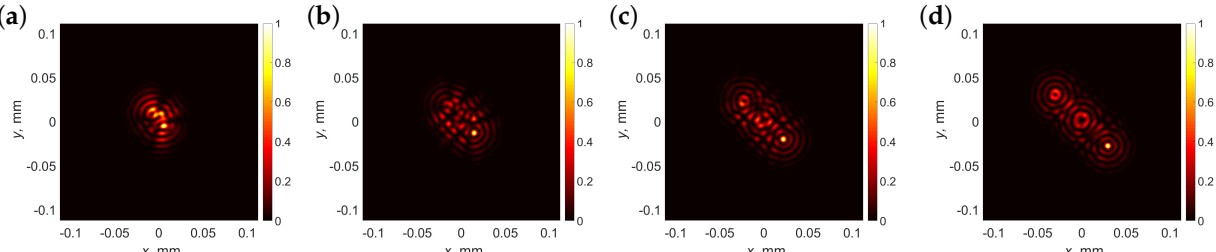

**Figure 7.** Transverse intensity distributions for an array of three optical needles with topological charges $m_1 = 2$, $m_2 = 1$, and $m_3 = 0$ and the length of the axial super-Gaussian profile $L = 2$ mm. The spatial separation of individual optical needles was $\rho_{12} = 20\lambda$ (**a**), $\rho_{12} = 40\lambda$ (**b**), $\rho_{12} = 60\lambda$ (**c**), and $\rho_{12} = 80\lambda$ (**d**).

As expected, the spatial separation $\rho_{12} = \rho_{13} = 20\lambda$ did not show improvement; see Figure 7a. The slight increase of $\rho_{12} = \rho_{13}$ to $40\lambda$ (see Figure 7b) demonstrated that only the non-vortical spike was recognized. However, most notably, when $\rho_{12} = \rho_{13} = 60\lambda$ (see Figure 7c), we observed that the topological charge needle $m = 2$ being central affected adjacent beams more than in the previous case, compared to Figure 6g. This means that the choice of the optimal separation distance depended not only on the individual topological charges, but also on the local position of that structure within an array. Lastly, we selected $\rho_{12} = \rho_{13} = 80\lambda$; see Figure 6d. When comparing this combined beam with the previous case (see Figure 6h), we observed that, due to the larger spatial extent, the vortical

needle with topological charge $m = 2$ was now more distorted due to the presence of the two adjacent optical needles.

### 3.5. Creation of an Array of Vortical Needles with Complex Positions and Axial Profiles

Lastly, given we found an optimal separation distance of $\rho_{12} = 80\lambda$, we aimed to create a more-complex array of vortical Bessel-like needles with individual positions in the space around the focal point; see Figure 8.

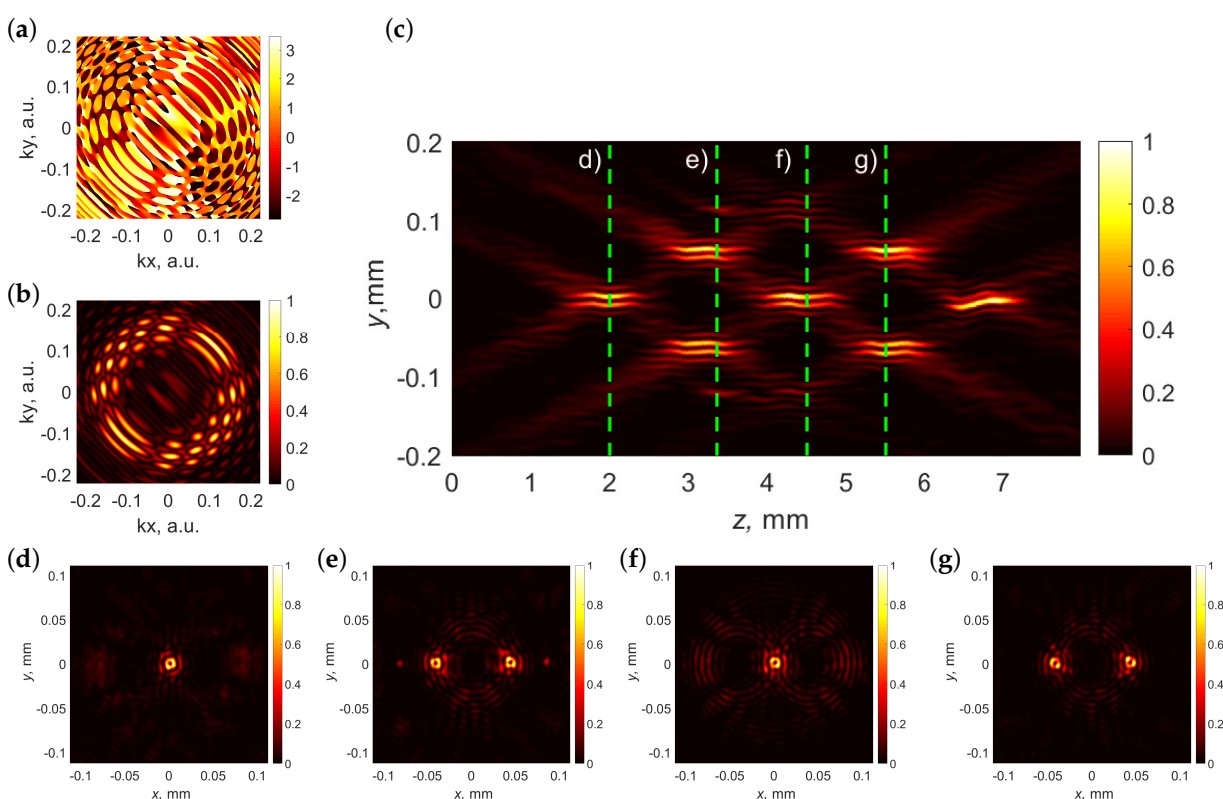

**Figure 8.** Phase (**a**) and amplitude (**b**) of the spatial spectra of a complex array of vortical needles. Longitudinal intensity distribution of an array of seven vortical optical needles with individual positions and topological charges (**c**). Transverse intensity distributions of an array and the particular positions in (**c**), marked green; see (**d**–**g**).

In this case, we created a structure of seven independent vortical-needle-like structures. For convenience, we selected their lengths to be the same, that is $L = 1$ mm and topological charges $m = 1$; however, the positions of those individual vortex needles were different.

The amplitude and phase distribution of the mask are shown in Figure 8a,b. The longitudinal intensity profile is shown in Figure 8c. We observed the intended propagation of the combined beam, although the vortical cores with topological charges $m = 1$ were not optimally resolved. This effect was caused by the fact that the resolution of the SLM in use was somehow pushed to the limit, and the $z$ scan quality, blur, appeared due to scan inhomogeneities and some inevitable inaccuracies. The signal-to-noise ratio in this case was estimated from Figure 8e comparing the minimal value in the center to the maximum value, and it was $SNR = 200$.

The inspection of the array at four individual positions $z$ is presented in Figure 8d–g. We observed some side lobes in parts peripheral to the vortex cores, but the main vortical pairs of vortical needles seemed to be intact, though slightly distorted.

## 4. Discussion and Conclusions

We discussed the creation of complex structures consisting of a number of individual vortical optical needles with individual topological charges.

We introduced a versatile method that empowers the controlled creation of arrays of parallel optical vortical needles with independent axial intensity profiles and topological charges. Our analysis delved into the interplay between the separation of individual optical vortical needles and their respective lengths. We observed that destructive interference between adjacent needles was less pronounced when they varied in length. The preliminary findings suggest that this phenomenon arose from the distinct spatial modulation in the Fourier space associated with optical needles of different lengths. The distortion between neighboring optical needles was the result of spatial overlap, emphasizing the importance of minimizing such overlap for optimal results. However, this technique imposes limitations on the smallest separation between the beams, the length of the needle, the topological charge, or the width of the beam.

Additionally, we introduced a step-like axial intensity profile described using a super-Gaussian function. This choice mitigated issues related to the Gibbs phenomenon, ensuring a smooth varying profile and sharp edges. Our experimentation identified the optimal parameter for the super-Gaussian function as $N = 7$.

In conclusion, the presented method facilitates the creation of diverse spatial intensity distributions in three dimensions, potentially finding applications in specific microfabrication tasks or other contexts.

**Author Contributions:** Conceptualization, S.O.; methodology, S.O.; software, S.O. and P.Š.; validation, P.Š.; formal analysis, S.O.; investigation, P.Š.; resources, S.O.; data curation, P.Š.; writing—original draft preparation, S.O.; writing—review and editing, S.O. and P.Š.; visualization, P.Š.; supervision, S.O.; project administration, S.O.; funding acquisition, S.O. All authors have read and agreed to the published version of the manuscript.

**Funding:** This research received funding from the Research Council of Lithuania (LMTLT) via agreement No. [S-MIP-23-71].

**Institutional Review Board Statement:** Not applicable.

**Informed Consent Statement:** Not applicable.

**Data Availability Statement:** The data will be made available upon request.

**Conflicts of Interest:** The authors declare no conflicts of interest.

## Abbreviations

The following abbreviations are used in this manuscript:

| | |
|---|---|
| SLM | spatial light modulator |
| FWHM | full-width at half-maximum |
| CCD | charge-coupled device |

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
