# Peer review of "Creating an Array of Parallel Vortical Optical Needles"

_photonics, doi:10.3390/photonics11030203_

Round 1

Reviewer 1 Report

Comments and Suggestions for Authors

In this paper, the authors suggest a method for creating Bessel-like optical needles with individual topological charges and adjustable individual axial intensity patterns. The results in this work are beneficial for various practical applications. The paper may be accepted for publication in the journal after getting the proper implementation of the following comments/suggestions.

1. I Suggest the authors rewrite the title, it is too long. 

2. The abstract section is not relevant, physically speaking. What are the new ideas and concepts compare to? I suggest the authors rewrite the abstract also.

3. Introduction: The authors provide a summary of works on the Bessel beam a non-diffractive beam, in my opinion, the authors have recently missed developments in the field. To give the readers a comprehensive overview of this promising field I suggest including the following references in the manuscript: Optica 10, 1161-1164 (2023), and Scientific Rep 12, 14064 (2022), in this paper authors demonstrated self healing Bessel beam by using an axicon and compare with a new non-diffractive space time light sheet and proved that this beam is speckle resistant. 

4. Figure 1: Can authors show the intensities distribution of the axial profiles show in fig. 1 for the simulations and or experimental measurements, readers can be interested.

5. Figure 3: Can authors explain why they use the Brewster polarizer? and also what are a 10x microscope objective, a collimating lens L5 (f = 200 mm) use for?

6. Authors mentioned in line 119 that "The optical setup used in this work is a bit different from that used in our previous research [19–21]" I can not see the different with fig 5. in ref 19, can authors give more details?

7. Figure 4: Why it has no hole in the Longitudinal axial profiles of vortical optical needles for the fig. 4 h and l, because of the hole in the central lobe. 

To my opinion, this manuscript can be recommended for publication in Photonics before the authors give the improvement.

Author Response

Responses to Reviewer 1

Reviewer 1.

In this paper, the authors suggest a method for creating Bessel-like optical needles with individual topological charges and adjustable individual axial intensity patterns. The results in this work are beneficial for various practical applications. The paper may be accepted for publication in the journal after getting the proper implementation of the following comments/suggestions.

Response: We thank the Reviewer for his/her insights and for supporting the publication of the work.

  1. I Suggest the authors rewrite the title, it is too long. 

Response: As suggested we have shortened title to “Creating An Array Of Parallel Vortical Optical Needles”

  1. The abstract section is not relevant, physically speaking. What are the new ideas and concepts compare to? I suggest the authors rewrite the abstract also.

Response: As suggested we have rewritten the abstract to address his/her suggestions

  1. Introduction: The authors provide a summary of works on the Bessel beam a non-diffractive beam, in my opinion, the authors have recently missed developments in the field. To give the readers a comprehensive overview of this promising field I suggest including the following references in the manuscript: Optica 10, 1161-1164 (2023), and Scientific Rep 12, 14064 (2022), in this paper authors demonstrated self healing Bessel beam by using an axicon and compare with a new non-diffractive space time light sheet and proved that this beam is speckle resistant. 

Response: We have significantly rewritten and extended the introduction section with more references related to the subject of beam shaping. 

  1. Figure 1: Can authors show the intensities distribution of the axial profiles show in fig. 1 for the simulations and or experimental measurements, readers can be interested.

Response: We show axial profiles of the beam having topological charge m=2 in fig. 4 (p).

  1. Figure 3: Can authors explain why they use the Brewster polarizer? and also what are a 10x microscope objective, a collimating lens L5 (f = 200 mm) use for?

Response: The polarizer serves 2 purposes: 1) together with a half-waveplate it attenuates the beam 2) ensures the beam sent to SLM has the polarization required for phase modulation. The vertical dimension of the SLM is 8.64 mm (we now include resolution and pitch in the manuscript). This is the maximum size of the ring that we can form. Demagnifying this 3x with our 4f setup gives 2.88mm ring. The Fourier lens is 24.5mm which results in NA of ~0.06. According to formula 2*2.405*NA/(2*pi*lambda) the size of the central spot would be 6.92 um. This would illuminate only a few pixels on the camera. Hence, the magnification is needed to resolve the beam. 

  1. Authors mentioned in line 119 that "The optical setup used in this work is a bit different from that used in our previous research [19–21]" I can not see the different with fig 5. in ref 19, can authors give more details?

Response: We now note in the manuscript that a different set of lenses is used this time. In the previous work magnification for the 4f imaging setup was ~6x now  it is 3x. 

  1. Figure 4: Why it has no hole in the Longitudinal axial profiles of vortical optical needles for the fig. 4 h and l, because of the hole in the central lobe. 

Response: We apologize for the confusion. Figures 4 h and l show intensity distribution at a single point of the azimuth on the first ring. In Figure 4 (p) we now show the intensity distribution cross section, where intensity minima can be clearly seen in the middle. We hope this clarifies the confusion. 

To my opinion, this manuscript can be recommended for publication in Photonics before the authors give the improvement.

Reviewer 2 Report

Comments and Suggestions for Authors

This is a good manuscript. I truly enjoyed reading this manuscript. I have found the manuscript to be scientifically sound, actual and important to the field in satisfactory form.

 As far as I know, in this paper, controllable spatial array of Bessel-like vortical optical needles with different topological charges and locations in the transverse plane is studied. A novel method for engineering Bessel vortices has been proposed. To verify the theoretical and numerical results,  controllable spatial arrays of individual beams has been generated, and the authors conduct a comprehensive study of the proposed method.

The title clearly identifies subject matter and the abstract is succinct, comprehensible to a non-specialist. The manuscript seems me clearly written and logically organized. Length is appropriate to topic. Quality of writing is adequate.

 The suggestions are as follows,

(i)In this paper, the abstract needs to be rewritten to make it richer.

(ii)In this paper, the introduction does not highlight the focus and innovation of this paper, so it is suggested to concentrate on the focus and innovation of the paper.

Author Response

Responses to Reviewers

Reviewer 2.

This is a good manuscript. I truly enjoyed reading this manuscript. I have found the manuscript to be scientifically sound, actual and important to the field in satisfactory form.

 As far as I know, in this paper, controllable spatial array of Bessel-like vortical optical needles with different topological charges and locations in the transverse plane is studied. A novel method for engineering Bessel vortices has been proposed. To verify the theoretical and numerical results,  controllable spatial arrays of individual beams has been generated, and the authors conduct a comprehensive study of the proposed method.

The title clearly identifies subject matter and the abstract is succinct, comprehensible to a non-specialist. The manuscript seems me clearly written and logically organized. Length is appropriate to topic. Quality of writing is adequate.

Response: We thank the Reviewer for his/her insights and for supporting the publication of the work.

 The suggestions are as follows,

(i)In this paper, the abstract needs to be rewritten to make it richer.

Response:As suggested by the Reviewer, we have rewritten the abstract to address his/her suggestions

(ii)In this paper, the introduction does not highlight the focus and innovation of this paper, so it is suggested to concentrate on the focus and innovation of the paper.

Response: We have significantly rewritten and extended the introduction section with more references related to the subject of beam shaping.

Reviewer 3 Report

Comments and Suggestions for Authors

The manuscript titled “Controllable spatial array of Bessel-like vortical optical needles with different topological charges and locations in the transverse plane” by Paulius Šlevas et al. presents simulation and experimental results on the engineering of Vortex Bessel beams. They have successfully generated array of Vortex Bessel beams using spatial light modulator. The results also include the axial profile engineering i.e. flat axial intensity. This will have significant applications in the field of material processing, plasma physics etc. Therefore, I recommend this manuscript for publication in the journal of Photonics although with the following suggestions. The authors may highlight the spatial beam characteristics such as cone angle, central core diameters of generated Vortex Bessel beams. If generated, authors may also include the results on Vortex axial intensity profile other than super-Gaussian type. Ref. 28 also seems to be incomplete.     

Author Response

Reviewer 3

The manuscript titled “Controllable spatial array of Bessel-like vortical optical needles with different topological charges and locations in the transverse plane” by Paulius Šlevas et al. presents simulation and experimental results on the engineering of Vortex Bessel beams. They have successfully generated array of Vortex Bessel beams using spatial light modulator. The results also include the axial profile engineering i.e. flat axial intensity. This will have significant applications in the field of material processing, plasma physics etc. Therefore, I recommend this manuscript for publication in the journal of Photonics although with the following suggestions. The authors may highlight the spatial beam characteristics such as cone angle, central core diameters of generated Vortex Bessel beams. If generated, authors may also include the results on Vortex axial intensity profile other than super-Gaussian type. Ref. 28 also seems to be incomplete.     

Response: We thank the Reviewer for his/her insights and for supporting the publication of the work. We have significantly rewritten and extended the introduction section with more references related to the subject of beam shaping. Appropriate changes to the discussion were introduced

Reviewer 4 Report

Comments and Suggestions for Authors

The authors report the generation of spatial array of Bessel like optical needles with several values of topological charges. The overall quality of the paper is good, thus including the theoretical calculations of a diversity of transverse and longitudinal profiles. The resulst are of interest and may contribute to applications. According to the present form I recommend to consider the following questions and comments :

- In the experimental part give details about the the properties of the LCOS to generate the phase patterns. Number of pixels - Phase levels - Required phase accuracy and stability for different values of the topologic charges. Limitations of the SLM properties for more complex array of optical needles. 

- The transverse intensity distributions in Fig 4 are clear but it is of interest to scan with accuracy some profiles along the x direction . Which size of the bright peak or dark hole. Comparaison with classical diffraction limit and quantify the intensity level of the side lobes. Also contribution of coherent noise to reduce the SNR on the longitudinal and transverse directions of the needles.

To conclude the paper brings significant new results on programmable array of optical needles. Both the theory and the experimental results are well introduced and developed. It is a valuable contribution for publication after taking account of the above questions and in the final form. 

Author Response

Responses to Reviewers

Reviewer 4.

The authors report the generation of spatial array of Bessel like optical needles with several values of topological charges. The overall quality of the paper is good, thus including the theoretical calculations of a diversity of transverse and longitudinal profiles. The resulst are of interest and may contribute to applications. According to the present form I recommend to consider the following questions and comments :

Response: We thank the Reviewer for his/her insights and for supporting the publication of the work.

- In the experimental part give details about the the properties of the LCOS to generate the phase patterns. Number of pixels - Phase levels - Required phase accuracy and stability for different values of the topologic charges. Limitations of the SLM properties for more complex array of optical needles. 

Response: We now include the number of pixels, pixel pitch and available phase bits in the manuscript. The SLM matrix encodes the phase using 8-bit information, so one has 256 levels of the phase. This accuracy was enough to test vortical needles with relatively high topological charges, though our aim never was hitting the limits imposed by the SLM. This interesting topic requires further investigations

- The transverse intensity distributions in Fig 4 are clear but it is of interest to scan with accuracy some profiles along the x direction . Which size of the bright peak or dark hole. Comparaison with classical diffraction limit and quantify the intensity level of the side lobes. Also contribution of coherent noise to reduce the SNR on the longitudinal and transverse directions of the needles.

Response: We provide an x cross section of m=2 beam intensity profiles in figures 4 (m-o). We now also note the sizes of the central peak and the central dark hole in the text. We now discuss the intensity levels of some of the side rings (case of m=2) and compare it to the ideal Bessel beam of the same topology.  

In our case f/# was approximately 9. This number gives following estimates https://www.edmundoptics.com/knowledge-center/application-notes/imaging/limitations-on-resolution-and-contrast-the-airy-disk

The Airy disk is 11.5 um which is approximately  22 wavelengths. In accordance with that we can’t resolve optical needles with separation distance of 20 lambdas, however the situation is better for cases with separation distance of 40 lambdas.

The SNR estimate was performed for a few experimental data and the SNR was found to be larger than 200.

To conclude the paper brings significant new results on programmable array of optical needles. Both the theory and the experimental results are well introduced and developed. It is a valuable contribution for publication after taking account of the above questions and in the final form.

Round 2

Reviewer 1 Report

Comments and Suggestions for Authors

The paper can be publish in the journal.